# A Double Cross-Linked Injectable Hydrogel Derived from Muscular Decellularized Matrix Promotes Myoblast Proliferation and Myogenic Differentiation

**DOI:** 10.3390/ma16155335

**Published:** 2023-07-29

**Authors:** Zhao Huang, Jianwen Cheng, Wei Su

**Affiliations:** Department of Orthopedics Trauma and Hand Surgery, The First Affiliated Hospital of Guangxi Medical University, Nanning 530021, China; huangz_gxmu@163.com (Z.H.); chengjwgxmu@163.com (J.C.)

**Keywords:** hydrogels, biomaterials, muscular decellularized matrix, proliferation, myogenic differentiation

## Abstract

Injectable hydrogels possess tremendous merits for use in muscle regeneration; however, they still lack intrinsic biological cues (such as the proliferation and differentiation of myogenic cells), thus considerably restricting their potential for therapeutic use. Herein, we developed a double cross-linked injectable hydrogel composed of methacrylamidated oxidized hyaluronic acid (MOHA) and muscular decellularized matrix (MDM). The chemical composition of the hydrogel was confirmed using ^1^H NMR and Fourier transform infrared spectroscopy. To achieve cross-linking, the aldehyde groups in MOHA were initially reacted with the amino groups in MDM through a Schiff-based reaction. This relatively weak cross-linking provided the MOHA/MDM hydrogel with satisfactory injectability. Furthermore, the methacrylation of MOHA facilitated a second cross-linking mechanism via UV irradiation, resulting in improved gelation ability, biomechanical properties, and swelling performance. When C2C12 myogenic cells were loaded into the hydrogel, our results showed that the addition of MDM significantly enhanced myoblast proliferation compared to the MOHA hydrogel, as demonstrated by live/dead staining and Cell Counting Kit-8 assay after seven days of in vitro cultivation. In addition, gene expression analysis using quantitative polymerase chain reaction indicated that the MOHA/MDM hydrogel promoted myogenic differentiation of C2C12 cells more effectively than the MOHA hydrogel, as evidenced by elevated expression levels of myogenin, troponin T, and MHC in the MOHA/MDM hydrogel group. Moreover, after four to eight weeks of implantation in a full-thickness abdominal wall-defect model, the MOHA/MDM hydrogel could promote the reconstruction and repair of functional skeletal muscle tissue with enhanced tetanic force and tensile strength. This study provides a new double cross-linked injectable hydrogel for use in muscular tissue engineering.

## 1. Introduction

Muscular tissue plays tremendous vital roles in the human body, such as maintaining human activities, connecting bones and joints, and protecting human tissues. Volumetric muscle loss frequently occurs in patients and can cause persistent dysfunction without effective repair [1]. The current conventional treatment for muscle loss is surgical reconstruction using autologous muscular flaps, but it faces several limitations, such as severe surgical trauma, low survival rates, donor site morbidity, and unavailability of donor sources [2]. Therefore, novel therapeutic strategies for the efficient treatment of muscle loss are urgently needed. The emerging technology of muscular tissue engineering using well-designed biomaterials holds promise in rehabilitating defective muscle [3].

The selection of the ideal biomaterial scaffold that can promote myoblast proliferation and myogenic differentiation is a crucial step in muscle tissue engineering [4,5]. Among various biomaterials used in muscular tissue engineering, injectable hydrogels show the most promising superiorities, which include the absence of trauma and surgical incisions, shortened skin healing time, and possible avoidance of wound infections [6], thus serving as fillers to help fill irregular defects and acting as carriers to deliver cells and growth factors to the target lesion [7]. However, most hydrogels are biologically inert and cannot promote cell proliferation or express myogenic markers [8].

Because myoblasts are accommodated within a muscle-specific microenvironment that is composed mainly of native extracellular matrix (ECM), exploitation of the features of muscle-specific ECM could be beneficial for muscle regeneration [9]. Consequently, endowing the injectable hydrogel with intrinsic muscular ECM components may boost muscle regeneration. However, since muscle-specific ECM comprises complex three-dimensional functional molecules, the synthesis of this material cannot be fully reproduced in the laboratory [10].

Previous studies revealed that improved cellular maturation, functioning, and tissue formation may be achieved using composite material scaffolds that combine topographical and electrically conductive cues to control skeletal muscle cell architecture [7]. However, the development of those composite scaffolds is sophisticated and complicated. Some studies indicated that the addition of bioactive factors could significantly promote muscle regeneration. For example, insulin-like growth factor 1 (IGF-1) plays a role in muscle cell differentiation, myoblast proliferation, and survival [11]. However, the encapsulation of the bioactive factor is not cost-effective, and the development of a sustained release system poses a formidable knot.

The natural muscular matrix plays a crucial role in activating the resident stem cells in a dormant state to enter the cell division cycle and experience myogenesis, thus advancing the muscle repair and regeneration process [12]. In addition, the muscular matrix also contains several intrinsic bioactive factors to promote the proliferation of myoblast [13]. Hence, we conjectured that a muscular matrix-derived hydrogel could be a promising substitute for use in muscular tissue engineering.

To overcome the immunogenicity of the allogenic muscular matrix, the muscular matrix was decellularized to obtain a muscular decellularized matrix (MDM). Extensive studies revealed that MDM could preserve most of its original bioactivity components, thus retaining a muscle-specific microenvironment [12]. Specifically, the MDM retained several favorable active factors for muscle regeneration, such as the retained IGF-1 in MDM may possess pro-myogenic properties. Hence, MDM is regarded as a suitable reservoir for the loading of bioactive factors. Moreover, studies also suggest that the MDM’s biodegradable byproduct can prompt endogenous progenitor cells to migrate and multiply in a regenerative response that refills the host tissue [14], thus, making MDM a promising candidate in muscle tissue engineering. Consequently, in combination with MDM, it is rational to prepare an optimal injectable hydrogel for use in muscle regeneration.

Herein, we aimed to develop a double cross-linked injectable MOHA/MDM hydrogel based on methacrylamidated oxidized hyaluronic acid (MOHA) and MDM, as illustrated in Figure 1. It was further investigated whether the MOHA/MDM hydrogel promoted the proliferation and differentiation of myogenic cells both in vitro and in vivo, showing high potential for use in muscle regeneration.

## 2. Materials and Methods

### 2.1. MOHA Synthesis

HA (1 g, Sigma, St. Louis, MO, USA) with molecular weights of 490 kD was dissolved in deionized water (100 mL), and then 0.5 M NaIO_4_ (5 mL, Sigma, St. Louis, MO, USA) was dropwise added for 4 h in the dark. The unreacted NaIO_4_ was inactivated using ethylene glycol. Thereafter, the achieved solution was exhaustively dialyzed using deionized water for three days and freeze-dried to obtain oxidized HA (OHA). Then, OHA (1 g) was dissolved in deionized water (DW, 100 mL), and 1 mL of methacrylic anhydride (molecular weight: 154.16, purity: >94%, Cas: No. 760-93-0, Sigma, St. Louis, MO, USA) was added to react for 12 h at −20 °C and pH = 8~8.5. To create MOHA, the solution was dialyzed for two days and then freeze-dried.

By counting the number of aldehyde groups in the polymer using t-butyl carbazate, the percentage of oxidized HA was determined. In brief, an excess amount of t-butyl carbazate was added to the OHA solution, and the amount of unreacted carbazate was calculated by adding a trinitrobenzenesulfonic acid solution. Spectrophotometric quantification of the colored reaction product was performed at 334 nm. The number of aldehydes in the solution can be calculated by subtracting the total number of unreacted carbazates from the total number of added carbazates.

The degree of methacrylation was verified via ^1^H NMR spectroscopy. Briefly, MOHA was dissolved in heavy water (D_2_O) at an 8 mg/mL concentration. A Varian Unity 300 spectrophotometer was used to record the spectra after NaCl was added to the solution to decrease its viscosity.

### 2.2. Preparation of the MOHA/MDM Hydrogel

The obtained muscle tissue derived from the rectus abdominis muscle (Shanghai Jiagan Breeding Factory, Shanghai, China) underwent a conventional decellularization protocol to achieve the MDM [15]. The muscle tissue was subjected to continuous agitation at a constant temperature shaker, with the ambient temperature set at 4 °C. Subsequently, it was washed in DW for a duration of 72 h. Following this, the tissue was lysed using a 4% sodium deoxycholate solution for a period of 4 h. Additionally, the lysate was subjected to digestion using 2000 kU DNase-I in the presence of 1 M NaCl for 3 h. This entire process was repeated twice in order to obtain the desired MDM.

The MOHA was dissolved in DW (5% *v*/*w*) and supplemented with 0.1% (*w*/*v*) 2-Hydroxy-4′-(2-hydroxyethoxy)-2-methylpropiophenone (I2959, Aladdin), which works as a photoinitiator. The MDM was dissolved in DW (5% *v*/*w*) and mixed with the MOHA solution mentioned above to initiate cross-linking by a Schiff-based reaction. It was subjected to further cross-linking by exposure to UV_365nm_ (10 mW/cm^2^) for 3 min, thus developing a double cross-linked injectable MOHA/MDM hydrogel.

A Fourier transform infrared spectrometer (FTIR, Nicolet is50, ThermoFisher, Waltham, MA, USA) was used to characterize the MOHA/MDM hydrogel. The injectability of the hydrogel before UV irradiation was investigated by manual injections using a 22-gauge needle. The gelation property of the hydrogel was examined after 3 min of UV irradiation.

The rheological experiments were carried out on three distinct groups: MOHA/MDM with UV irritation (referred to as MOHA/MDM + UV), MOHA with UV irritation (referred to as MOHA + UV), and MOHA/MDM without UV irritation (referred to as MOHA/MDM). To measure the viscosity of the hydrogel, an Anton Paar MCR302 rheometer with a parallel plate (25 mm diameter) configuration was employed. In brief, the hydrogels were applied to cover the bottom plate of the rheometer with parallel-plate geometry. Subsequently, the gap between the plates was set at 1 mm, and the rheometer was covered. The hydrogel viscosity was then measured at room temperature across shear rates ranging from 0.1 to 100 s^−1^, both with and without UV irritation. Following exposure to UV irradiation (365 nm, 10 mW/cm^2^) for 100 s or without exposure, dynamic rheology tests were conducted. These tests were performed at a fixed frequency of 6 rad/s and a strain of 1%, with the purpose of determining the storage modulus (G′) of the hydrogels.

Using a Nanoindenter (Piuma Chiaro, OPTICS II, Amsterdam, The Netherlands) with a falling speed of 10 µm/s, Young’s modulus of samples in MOHA/MDM + UV, MOHA + UV, and MOHA/MDM groups were determined.

The swelling ratio of samples in MOHA/MDM + UV, MOHA + UV, and MOHA/MDM groups was evaluated using a gravimetrical method. Briefly, hydrogels were initially dried and weighted as α. Thereafter, the dried samples were immersed in DW (20 mL). At scheduled time points (1, 2, 3, 4, 6, 12, 18, 24 h), the samples were retrieved from the DW, blotted with napkin tissue, and weighted as β. The swelling ratio was calculated as follows: (β − α)/α × 100%.

### 2.3. Proliferation of Myoblasts in Hydrogels

C2C12 myogenic cells, procured from the Chinese Academy of Sciences’ Shanghai Cell Bank (Shanghai, China), were suspended within the MOHA/MDM and MOHA hydrogels precursor at a concentration of 1 × 10^6^ cells/mL. Thereafter, the cell-loaded hydrogel was gelated using UV light and incubated in Dulbecco’s modified Eagle’s medium (Gibco, Billings, MT, USA) supplemented with 10% fetal bovine serum (Gibco, Billings, MT, USA) and 1% penicillin-streptomycin (Gibco, Billings, MT, USA). We also established a control group in which identical cells were cultured in the Petri dish. The viability of the cells in the hydrogels was evaluated by staining using the live/dead cell viability assay (Invitrogen, Waltham, MA, USA) and examined under a confocal laser scanning microscope (FV1200; Olympus, Tokyo, Japan) after one, four, and seven days of in vitro culture. The live and dead cell numbers were counted from the achieved images. Cell proliferation was also measured using a cell counting kit-8 (CCK-8; Dojindo, Kumamoto, Japan) following the manufacturer’s instructions, and optical density (OD) was measured at 450 nm.

### 2.4. Myogenic Differentiation of Myoblasts in Hydrogels

Following seven days of in vitro culture as described in Section 2.2, the myoblast-loaded MOHA-AMDM hydrogel was further tested using a quantitative polymerase chain reaction (qPCR) to identify myogenic differentiation of myoblasts in the MOHA-AMDM hydrogel. The comparative threshold cycle approach was used to examine the data, and the results were then normalized to the endogenous reference gene β-actin. The primer sequences used are listed below in Table 1.

### 2.5. Muscle Repair in a Full-Thickness Abdominal Wall Defect Model

The protocol was approved by the Ethics Committee of Guangxi Medical University (No. 202204013, Nanning, China). Twelve male rats in total, purchased from Shanghai Jiagan Breeding Factory (Shanghai, China), were implanted with either MOHA/MDM or MOHA after being randomly classified into two groups (*n* = 6 for each group). Then, 10% chloral hydrate (4 mL/kg) was intraperitoneally injected into the rats to induce anesthesia. A full-thickness abdominal wall injury (2 × 2 cm^2^) involving the fascia, the underlying rectus abdominis muscle, and the peritoneum was created with a longitudinal midline skin incision of 2 cm. A 0.3 cm overlap was used when inserting the hydrogel intraabdominally, and an interrupted suture was used to secure it tension-free to the abdominal wall. After closing the skin incision, rats were allowed to recuperate normally. The rats were euthanized at four and eight weeks for the evaluation of muscle regeneration.

### 2.6. Histopathological Analysis

To observe muscular regeneration, the above samples were embedded, sectioned, treated with hematoxylin-eosin (HE) stain and Masson’s trichrome stains, and observed under an optical microscope. For immunohistochemical labeling, samples were incubated with the primary antibody against the cluster of differentiation desmin (1:400; Abcam, Cambridge, UK).

### 2.7. Tetanic Force Evaluation

The repaired samples with the hydrogel placement site covered by no more than 2 mm of surrounding muscle were retrieved eight weeks after implantation. A silk suture was used to attach the distal tissue to a force transducer (AD Instruments, Dunedin, New Zealand). Parallel platinum electrodes were positioned at the anastomoses near the implant site perpendicular to the direction of muscle contraction, which was determined by the nylon sutures. A model S48 stimulator was then used to apply electrical field stimulation (20 V at the electrodes) to the muscle (Grass Technologies, Middleton, WI, USA). Following a 10 min equilibration period, the optimal length was found based on the twitch response by varying the muscle’s length by rotating the micrometer head. A train of 0.2 ms square pulses lasting 1200 ms was used to measure the peak isometric contractile force at the optimal length over a range of frequencies (0–200 Hz). Power Lab/8sp (AD Instruments) was used to display and record force readings in real-time.

### 2.8. Biomechanical Analysis

Using a uniaxial material testing device (Instron5542, Boston, MA, USA), the biomechanics of 50 × 10 mm^2^ repaired samples four and eight weeks after implantation were examined. Briefly, samples were submerged in 100 mL PBS for one hour at 37 °C. After that, with the two ends of the sample held between the grippers, the distance between the grippers was set at 25 mm. Up until failure, samples were gradually stretched by enlarging the space between the grippers at a crosshead speed of 10 mm/min. The tensile strength was calculated from the achieved stress–strain curve.

### 2.9. Statistical Analyses

The data are presented as the mean ± standard deviation (SD), and each experiment was independently conducted at least three times. One-way ANOVA was used for comparisons between multiple groups, while Student’s *t*-test was used for comparisons between just two groups. All the statistical analyses were performed using GraphPad Prism 8 software. The level of statistical significance was set at * *p* < 0.05.

## 3. Results and Discussion

Hydrogels have attracted considerable interest as scaffolds for muscle regeneration due to their exceptional biocompatibility, low immunogenicity, plasticity, and resemblance to the three-dimensional porous extracellular matrix [16,17]. Compared to traditional scaffolds, hydrogels offer unique advantages, such as the ability to be administered via injection, facilitating minimally invasive procedures for accessing challenging sites. Additionally, their self-healing properties promote integration with host muscle tissue, making them suitable for treating large or irregular muscle defects [18]. However, the development of an ideal injectable hydrogel that can effectively promote myoblast proliferation and myogenic differentiation remains a challenge. MDM contains numerous ECM proteins and growth factors believed to be crucial for muscle regeneration [19]. However, there has been limited research focused on the development of an injectable hydrogel composed of MDM. In this study, we developed a double cross-linked injectable hydrogel using MOHA and MDM. The MDM was incorporated into the hydrogel through a Schiff-based reaction with the aldehyde-based MOHA, followed by UV cross-linking to further strengthen the gel.

Schiff-based reactions are commonly employed to create amide bonds, which have strong adhesive strength and are biocompatible [20]. It is well known that the natural ECM contains inherent amino groups [21], which could react with the aldehyde group via a Schiff-based reaction to form a cross-link. N-acetyl-D-glucosamine and D-glucuronic acid are repeating units that make up hyaluronic acid (HA), the molecular framework of proteoglycan complexes. It has cis-diol groups with carbon–carbon bonds, which are easily oxidized to produce reactive aldehyde groups that can be combined to form amide bonds by the Schiff-based reaction [22]. HA is frequently used for tissue engineering owing to its high biocompatibility and bioactive characteristics, which are known to enhance the proliferation of several cells and reduce inflammation [23,24]. In this study, HA was initially oxidized by NaIO_4_. Then, it was modified by methacrylation to prepare the MOHA hydrogel. Prior to ^1^H-NMR measurement, the aldehyde polymers were treated with t-butyl carbazate to analyze the aldehyde groups generated by the oxidation reaction. This demonstrated that MOHA had a 35% oxidation level. Two additional peaks at 5.0 ppm and 5.1 ppm were observed on the ^1^H NMR pattern of MOHA and OHA in each of the aldehyde-modified polysaccharides. These peaks were the hemiacetalic protons produced from the aldehyde groups and nearby hydroxyl groups (Figure 2A), indicating the successful synthesis of oxidated HA (OHA). A new peak at 5.7 ppm was observed on the ^1^H NMR pattern of MOHA, corresponding to the protons within the C=C bond of methacrylate. The ^1^H NMR pattern of MOHA further revealed that the methacrylate modification levels were approximately 25% (Figure 2A).

Thereafter, MOHA and MDM underwent a Schiff-based reaction and UV cross-linking to develop a double cross-linked injectable MOHA/MDM hydrogel. The FTIR spectrum of MOHA revealed the stretching band of the C=C bond at 1700 cm^−1^, confirming the successful methacrylate modification (Figure 2B). Moreover, the FTIR spectrum revealed that the MOHA/MDM hydrogel simultaneously contained specific peaks in MDM and MOHA, suggesting the successful combination of MOHA and MDM. In addition, the MOHA/MDM hydrogel before UV irradiation could be injected easily via needles and formed a continuous line, whereas the only MOHA hydrogel formed intermittent lines (Figure 2C), suggesting that the initial Schiff-based reaction in MOHA/MDM hydrogel is relatively weakly cross-linked and thus endows the MOHA/MDM hydrogel with satisfactory injectability. Moreover, our data revealed that both the MOHA/MDM and the MOHA hydrogels could readily undergo gelation after UV irradiation for 3 min (Figure 2D), owing to the successful methacrylate modification of HA.

The rheology tests further suggested that hydrogel in MOHA/MDM + UV group exhibited higher viscosity compared to that in MOHA + UV and MOHA/MDM groups (Figure 3A), and the storage modulus (G′) and Young’s modulus of hydrogel in MOHA/MDM + UV group were also higher than those in MOHA + UV and MOHA/MDM groups (Figure 3B,C), which were all attributed to the cross-link after Schiff-based reaction in MOHA/MDM hydrogel. Notably, considering the levels of viscosity, G′, and Young’s modulus in hydrogels present a trend of MOHA/MDM + UV > MOHA + UV > MOHA/MDM, corroborating the two-step (Schiff-based reaction + UV irradiation) cross-linking mentioned in the study. In addition, the viscosity of the hydrogels decreased, and G′ of hydrogels increased with increasing UV irradiation, indicating that gelation occurred after UV triggered the cross-linking and the rheology of hydrogel could be controlled by varying the irradiation time. The swelling ratio in all the MOHA/MDM + UV, MOHA + UV, and MOHA/MDM hydrogels increased between hours 0 and 12 of incubation in PBS and reached a plateau after 12 h of incubation (Figure 3D). The final swelling ratio in MOHA + UV hydrogel was approximately 60%, whereas that in MOHA/MDM + UV hydrogel was approximately 20%, suggesting that the Schiff-based reaction significantly promoted the shape maintenance of MOHA/MDM hydrogel. Notably, the final swelling ratio of MOHA/MDM hydrogel was approximately 70%, which was far higher than that in MOHA/MDM + UV hydrogel, demonstrating that UV irradiation could significantly strengthen cross-linking. Collectively, our data demonstrate that the first cross-link via Schiff-based reaction imparts the MOHA/MDM hydrogel with its favorable injectability, enhanced biomechanical properties, and shape maintenance, whereas the second cross-linking via UV irradiation endows the MOHA/MDM hydrogel with gratifying gelling property.

C2C12 is a multipotent cell line derived from muscle satellite cells that fuse with each other during differentiation to form myotubes in vitro. In this study, C2C12 cells were co-cultured with the MOHA/MDM hydrogel in vitro for seven days. Our results demonstrated that the MOHA/MDM hydrogel significantly promoted C2C12 proliferation compared with the MOHA counterpart, as evidenced by intensive live myoblast in live/dead staining images and fewer dead cells in the MOHA/MDM group than in the MOHA and control groups (Figure 4A–E). Notably, the images of live/dead staining in MOHA/MDM hydrogel are closer to spike-like morphology, whereas those in MOHA and control groups are in an almost round shape. The typical morphology of a myocyte is characterized by a long and narrow spindle shape. In contrast, C2C12 cells exhibit a round shape. Therefore, for C2C12 cells to undergo myogenic differentiation and become mature myocytes, it is imperative that they undergo a transformation from a round shape to a spindle shape. Consequently, the observed change in the morphology of C2C12 cells, from a round shape to a spike-like structure in the MOHA/MDM hydrogel, indicates the occurrence of myogenic differentiation.

Moreover, the CCK-8 assay also confirmed that the OD values in the MOHA/MDM group were significantly higher than those in the MOHA group (Figure 4F), indicating that the MOHA-AMDM hydrogel could promote myoblast proliferation.

By means of qPCR analysis, we were able to further quantify the gene expression levels of myogenesis-specific markers (myogenin, troponin T, and MHC). Muscle-specific transcription factor myogenin is expressed prior to the onset of the postmitotic stage. MHC and Troponin T are major components of the contractile muscle fibers. A common indicator of the efficiency and differentiation of in vitro-grown myotubes is the expression of MHC isoforms. Our data revealed that all three genes were expressed at a higher level in the MOHA/MDM + C2C12 cells group than in the MOHA + C2C12 cells, pure MOHA/MDM, and MOHA groups (Figure 5A–C), demonstrating that the MOHA/MDM hydrogel could enhance myogenic differentiation.

Histopathological evaluation is the gold standard in the diagnosis of myogenesis. The general HE staining revealed that more myofiber-specific structure was visible in MOHA/MDM group than that in the MOHA group at both four and eight weeks (Figure 6A). In addition, Masson’s trichrome staining further confirmed the enhanced myofiber deposition in MOHA/MDM group than that in just the MOHA group, as indicated by the red-stained myofiber (Figure 6B). Moreover, the positive myofiber expressions increased with increased implantation time. One of the last myogenesis markers, desmin, is essential for the stability of newly formed myocytes [25]. Hence, the expression of desmin is a sign of myogenesis. The immunohistochemical desmin staining revealed that desmin-positive myofibers increased with the increased implantation course from four to eight weeks, and the desmin-positive myofibers were considerably more intensive in MOHA/MDM group than that in the MOHA group (Figure 7A), further corroborating that the MOHA/MDM hydrogel promotes myogenic occurrence.

Rat rectus abdominis responses to electrical stimulation-induced tetanic force were used to measure the myogenesis in repaired tissue in the MOHA/MDM and MDM groups at eight weeks post-implantation (Figure 7B). Consistent with the histological and immunochemical findings, a higher tetanic force was observed for the repaired tissue in MOHA/MDM group than in the MDM group, suggesting a better functional recovery in the MOHA/MDM hydrogel. The in vivo biomechanical properties of repaired tissue in MOHA/MDM and MDM hydrogels were analyzed after four and eight weeks. The tensile strength in both hydrogels increased as the implantation course was prolonged, whereas the levels in MOHA/MDM group were significantly higher than those in the MOHA group at both four and eight weeks (Figure 7C).

The muscular matrix can be isolated from resident cells by decellularization strategies to result in an MDM material, serving as a non-immunogenic bioscaffold [26]. Recently, extensive studies revealed that MDM has good biocompatibility and negligible immunogenicity and could retain the regenerative and differentiated microenvironment for muscle regeneration [27]. One potential mechanism for the elevated capacities in proliferation and myogenic differentiation may be the retention of specific muscle ECM proteins, including collagens, elastin, adhesion proteins, and proteoglycans. The precise control of self-replication, proliferation, and differentiation modes is made possible by these insoluble components. The type of ECM receptor or integrin that a cell expresses is determined by its microenvironment. These receptors are expressed by cells, which bind to the appropriate ECM to enhance cell growth or differentiation [28]. In addition, the vast majority of endogenously generated growth factors that influence cell replication and differentiation state are sequestered by MDM. Moreover, the hydrogel provides a three-dimensional culture condition, which is a potent promyogenic cue, thus facilitating myogenic differentiation [29].

## 4. Conclusions

In this study, we focused on the development of a double cross-linked injectable hydrogel consisting of MOHA and MDM. The MOHA/MDM hydrogel undergoes a Schiff-based reaction upon injection, followed by cross-linking under UV light irradiation. Our results showed that the MOHA/MDM hydrogel surpassed the performance of the MOHA hydrogel in terms of injectability, rheological properties, biomechanics, and swelling behavior. Furthermore, our in vitro and in vivo experiments confirmed that the addition of MDM in the hydrogel formulation resulted in improved myoblast proliferation and myogenic differentiation compared to the MOHA hydrogel. Overall, this study presents a novel double cross-linked injectable hydrogel, combining MOHA and MDM, which holds great promise for applications in muscular tissue engineering.

## Figures and Tables

**Figure 1 materials-16-05335-f001:**
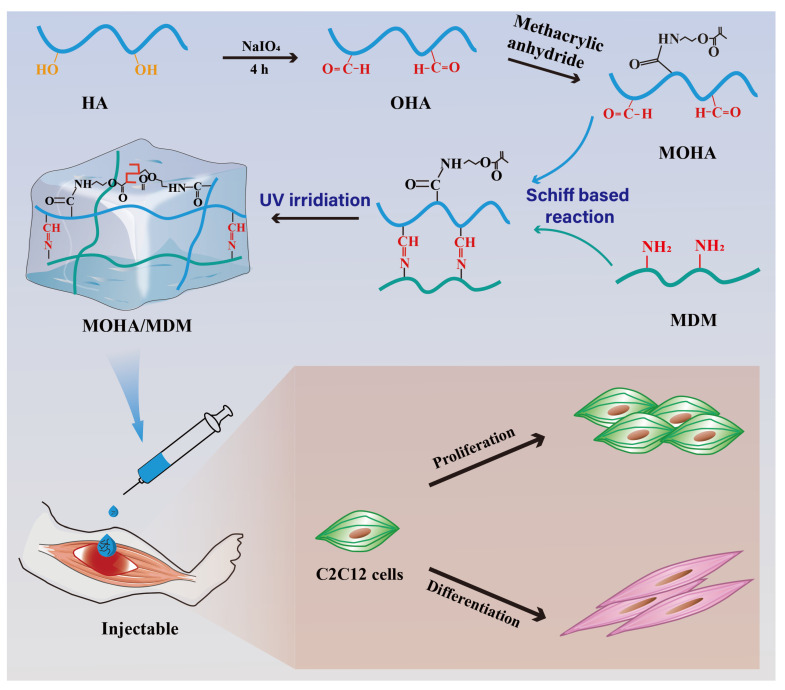
Schematic illustration for the development of MOHA/MDM hydrogel. Briefly, HA was oxidized by NaIO_4_ and methacrylated to prepare MOHA. Thereafter, MOHA and MDM underwent a Schiff-based reaction and UV cross-linking to develop a double cross-linked injectable MOHA/MDM hydrogel, aiming to simultaneously promote myoblast proliferation and myogenic differentiation.

**Figure 2 materials-16-05335-f002:**
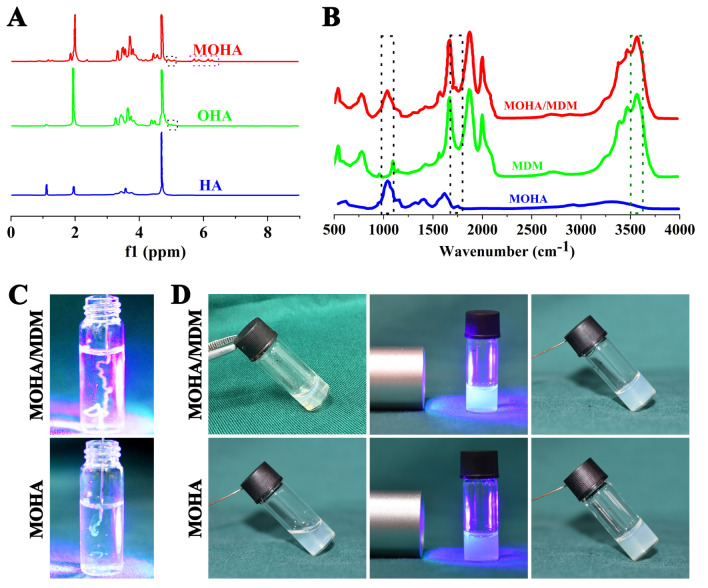
Preparation of the injectable MOHA/MDM hydrogel. The ^1^H NMR pattern of HA, OHA, and MOHA (**A**). The black squares outline specific peaks in OHA, whereas the purple square outlines specific peaks in MOHA. The FTIR spectra in the MOHA, MDM, and MOHA/MDM groups (**B**). The black squares outline specific peaks in MOHA, whereas the green square outlines specific peaks in MDM. The injectability of MOHA/MDM and MOHA hydrogels before UV irradiation (**C**). The gelation of MOHA/MDM and MOHA hydrogels via UV irradiation (**D**).

**Figure 3 materials-16-05335-f003:**
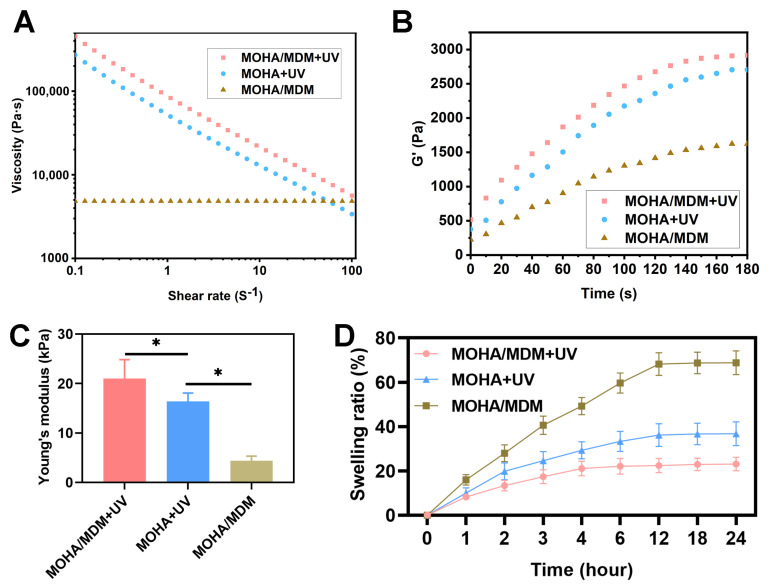
Characterizations of MOHA/MDM hydrogel. Viscosity depending on shear rate in MOHA and MOHA/MDM hydrogels with UV light at varying times and MOHA/MDM hydrogel without UV irritation (**A**). Dynamic storage modulus (G′) of MOHA and MOHA/MDM hydrogels with UV light at varying times and MOHA/MDM hydrogel without UV irritation (**B**). Young’s modulus of MOHA and MOHA/MDM hydrogels after 3 min UV irradiation and MOHA/MDM hydrogel without UV irritation (**C**). Swelling ratio of MOHA and MOHA/MDM hydrogels after 3 min UV irradiation and MOHA/MDM hydrogel without UV irritation from 0 to 24 h (**D**). * *p* < 0.05.

**Figure 4 materials-16-05335-f004:**
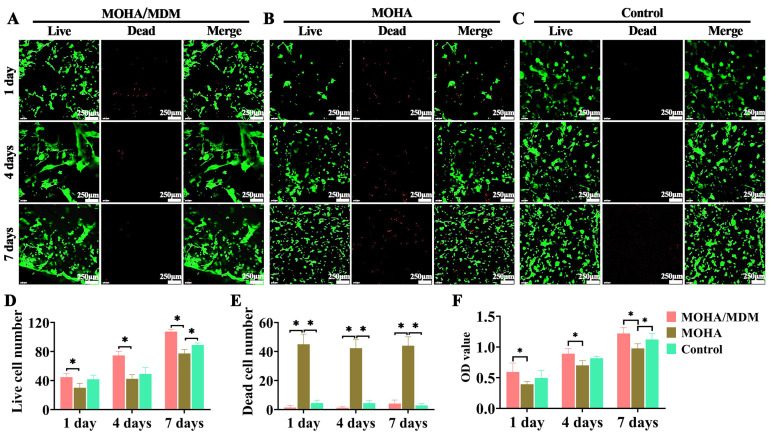
Myoblast proliferation within hydrogels after seven days of in vitro culture. Live/dead staining for the MOHA/MDM (**A**), MOHA (**B**), and control (**C**) groups. Quantitative data of live cell number (**D**), dead cell number (**E**), and OD value via CCK8 assay (**F**). * *p* < 0.05.

**Figure 5 materials-16-05335-f005:**
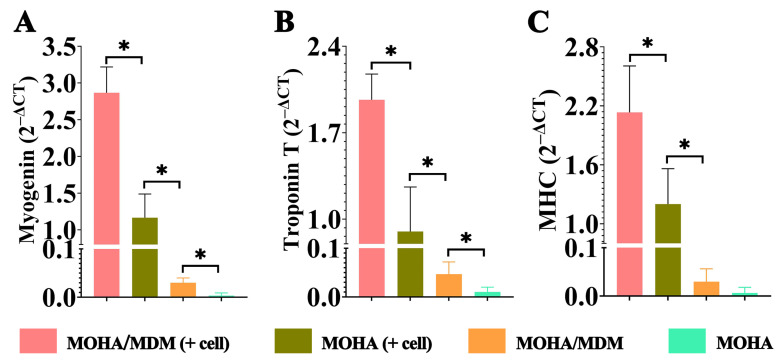
Myogenic differentiation of hydrogels with or without cells after seven days of in vitro culture. Genes expression of myogenin (**A**), troponin T (**B**), and MHC (**C**) in MOHA/MDM + cells, MOHA + cells, MOHA/MDM, and MOHA groups via qPCR examination. * *p* < 0.05.

**Figure 6 materials-16-05335-f006:**
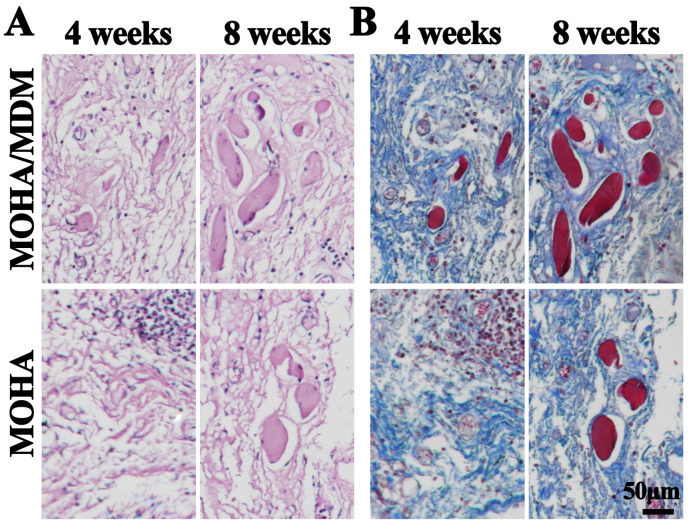
Myogenesis expression after in vivo implantation of MOHA/MDM and MOHA groups for four and eight weeks. HE (**A**) and Masson’s trichrome (**B**) staining of repaired tissue in MOHA/MDM and MOHA groups.

**Figure 7 materials-16-05335-f007:**
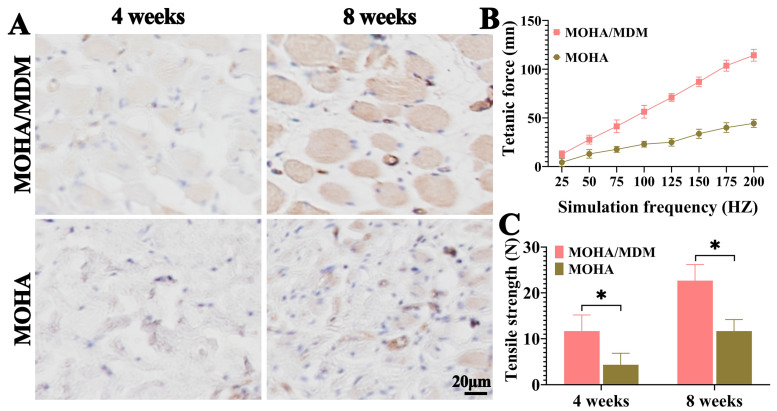
Immunohistochemical, electrobiological, and biomechanical evaluations of repaired tissue in MOHA/MDM and MOHA groups. Immunohistochemical desmin staining of repaired tissue in MOHA/MDM and MOHA groups at four and eight weeks (**A**). Tetanic force of repaired tissue in both groups at eight weeks (**B**). Tensile strength of repaired tissue in both groups at four and eight weeks (**C**). * *p* < 0.05.

**Table 1 materials-16-05335-t001:** Primers used in the qPCR reaction.

Genes	Sequence
Myogenin	forward: 5′ CTGACCCTACAGACGCCCAC 3′
	reverse: 5′ TGTCCACGATGGACGTAAGG 3′
Troponin T	forward: 5′ TCAATGTGCTCTACAACCGCA 3′
	reverse: 5′ ACCCTTCCCAGCCCCC 3′
MHC	forward: 5′ AGCAGACGGAGAGGAGCAGGAAG 3′
	reverse: 5′ CTTCAGCTCCTCCGCCATCATG 3′
β-actin	forward: 5′ AAGGAAGGCTGGAAAAGAGC 3′
	reverse: 5′ GCTACAGCTTCACCACCACA 3′

## Data Availability

The data that support the findings of this study are available from the corresponding author upon reasonable request.

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
