# Peer review of "A Double Cross-Linked Injectable Hydrogel Derived from Muscular Decellularized Matrix Promotes Myoblast Proliferation and Myogenic Differentiation"

_materials, 2023, doi:10.3390/ma16155335_

Round 1

Reviewer 1 Report

Ø  In the study, the authors kept the UV radiation time to only 3 minutes. Did the authors examine the properties of the sample exposed to radiation for more than 3 minutes, such as modulus or decay?

Ø  If I understood correctly, injectability in the study was tested before UV radiation, but what about after? Is it still injectable?

Ø  My advice to the authors is that the media used (DMEM) should be written clearly and the brand should be specified.

ؠ Why do the viscosity values ​​decrease while the measured G' values ​​increase under UV radiation?

Ø  The mechanism and contribution of species to crosslinking can be elucidated if the two-step (Schiff base reaction + UV crosslinking) crosslinking mentioned in the study, in which crosslinking (with UV radiation) is demonstrated under rheometer, can be demonstrated as a two-step modulus increase. I strongly advise authors to repeat this measurement.

Ø  Figure 2c figure caption injectability of gels etc. should be corrected.

Ø  Figure 3a figure cap change of viscosity depending on shear rate etc. should be corrected.

Ø  I ask the authors to zoom in on the y-scale in the graphs for Figures 3a and b.

Ø  Please change the unit of the y scale from KPA to kPa for Figure 3c.

Ø  I think it will appeal to the readers if the authors at least add the MDM protocol to the supplementary.

Ø  How do the authors explain the relationship between E and G? Young modulus is 3 kPa while shear modulus is around 80 kPa??

Ø  What do the authors think about the shape change of C2C12 cells in MOHA/MDM hydrogel?

Author Response

Detailed Responses to Reviewer 1:

Recommendation: Minor Revision

General Observation

In the study, the authors examined a new double-cross-linked injectable MOHA/MDM

hydrogel based on methacrylamidated oxidized hyaluronic acid (MOHA) and muscular

decellularized matrix (MDM) for use in muscular tissue engineering. MOHA hydrogel shows more efficient injectability, mechanical performance and swelling properties compared to MOHA/MDM hydrogel.

You can find my detailed opinions and suggestions about the study below. I think that this text can be published after the minor changes I suggested.

Response: Thank you for your professional comments. We have amended our work according to your useful suggestion, thus resulting in a considerable improvement of our manuscript. And the revised text was highlighted in red.

Minor revisions:

In the study, the authors kept the UV radiation time to only 3 minutes. Did the authors examine the properties of the sample exposed to radiation for more than 3 minutes, such as modulus or decay?

Response: Thank you for your comment. We have examined the properties of the sample exposed to radiation for more than 3 minutes, such as 5 min. Our data revealed that the modulus in 5 min were keep almost unchanged as in 3 min (shown as the following figure), which may ascribe to all the chemical bond used for cross-linking have been occupied.

Figure caption: Young's modulus for the MOHA/MDM hydrogel at 3, 4, and 5 min after UV irradiation.

Ø If I understood correctly, injectability in the study was tested before UV radiation, but what about after? Is it still injectable?

Response: Thank you for your comment. It correctly that the injectability was tested before UV radiation. After the UV radiation, the hydrogel was highly gelated, thus was almost not injectable.

Ø My advice to the authors is that the media used (DMEM) should be written clearly and the brand should be specified.

Response: Thank you for your useful suggestion. We have added the specific media and brand in our revised manuscript. Please note the revised text was highlighted in red.

Ø Why do the viscosity values ​​decrease while the measured G' values ​​increase under UV radiation?

Response: Thank you for your careful observation. This is because our color markings are incorrect. We feel sorry for this error, and we have corrected it in the revised manuscript of Figure 3.

Ø The mechanism and contribution of species to crosslinking can be elucidated if the two-step (Schiff base reaction + UV crosslinking) crosslinking mentioned in the study, in which crosslinking (with UV radiation) is demonstrated under rheometer, can be demonstrated as a two-step modulus increase. I strongly advise authors to repeat this measurement.

Response: Thank you for your professional comment and kindly suggestion. Considering our rheometer cannot measure while illuminating light in real-time, so we added a control group of MOHA/MDM without UV irritation. Consequently, the rheological experiments were carried out on three distinct groups: MOHA/MDM with UV irritation (referred to as MOHA/MDM + UV), MOHA with UV irritation (referred to as MOHA + UV), and MOHA/MDM without UV irritation (referred to as MOHA/MDM). And the updated results were displayed in the revised Figure 3 of our new version (shown in the following figure). Our data revealed that the levels of viscosity, G', and Young's modulus in hydrogels present a trend of MOHA/MDM + UV > MOHA +UV > MOHA/MDM, corroborating the two-step (Schiff-based reaction + UV irradiation) crosslinking mentioned in the study.

Figure caption: Characterizations of MOHA/MDM hydrogel. Viscosity depending on shear rate in MOHA and MOHA/MDM hydrogels with UV light at varying times and MOHA/MDM hydrogel without UV irritation (A). Dynamic storage modulus (G') of MOHA and MOHA/MDM hydrogels with UV light at varying times and MOHA/MDM hydrogel without UV irritation (B). Young's modulus of MOHA and MOHA/MDM hydrogels after 3 min UV irradiation and MOHA/MDM hydrogel without UV irritation (C). Swelling ratio of MOHA and MOHA/MDM hydrogels after 3 min UV irradiation and MOHA/MDM hydrogel without UV irritation from 0-24 h (D). *P < 0.05.

Ø Figure 2c figure caption injectability of gels etc. should be corrected.

Response: Thank you for your careful observation. We have corrected the figure caption of Figure 2c as: “The injectability of MOHA/MDM and MOHA hydrogels before UV irradiation (C).” Please note the revised text was highlighted in red.

Ø Figure 3a figure cap change of viscosity depending on shear rate etc. should be corrected.

Response: Thank you for your careful observation. We have corrected the figure caption of Figure 3a as: “Viscosity depending on shear rate in MOHA and MOHA/MDM hydrogels with UV light at varying times and MOHA/MDM hydrogel without UV irritation (A).” Please note the revised text was highlighted in red.

Ø I ask the authors to zoom in on the y-scale in the graphs for Figures 3a and b.

Response: Thank you for your careful observation. We have zoomed the y-scale in the graphs for Figures 3a and b.

Ø Please change the unit of the y scale from KPA to kPa for Figure 3c.

Response: Thank you for your careful observation. We have corrected this error in the revised Figure 3c of our revised version.

Ø I think it will appeal to the readers if the authors at least add the MDM protocol to the supplementary.

Response: Thank you for your careful observation. We have added the MDM protocol in our revised version: “The muscle tissue was subjected to a continuous agitation at a constant temperature shaker, with the ambient temperature set at 4 ℃. Subsequently, it was washed in DW for a duration of 72 h. Following this, the tissue was lysed using a 4% sodium deoxycholate solution for a period of 4 h. Additionally, the lysate was subjected to digestion using 2000 kU DNase-I in the presence of 1 M NaCl for 3 h. This entire process was repeated twice in order to obtain the desired MDM.”

Please note the revised text was highlighted in red.

Ø How do the authors explain the relationship between E and G? Young modulus is 3 kPa while shear modulus is around 80 kPa??

Response: Thank you for your careful observation. We have reconducted this experiment for 5 times and updated our data in the Figure 3c of our revised version. We feel sorry for this error.

Ø What do the authors think about the shape change of C2C12 cells in MOHA/MDM hydrogel?

Response: Thank you for your careful observation. The typical morphology of a myocyte is characterized by a long and narrow spindle shape. In contrast, C2C12 cells exhibit a round shape. Therefore, for C2C12 cells to undergo myogenic differentiation and become mature myocytes, it is imperative that they undergo a transformation from a round shape to a spindle shape. Consequently, the observed change in the morphology of C2C12 cells, from a round shape to a spike-like structure in the MOHA/MDM hydrogel, indicates the occurrence of myogenic differentiation.

Reviewer 2 Report

The manuscript described the synthesize and characterization of double cross-linked injectable MOHA/MDM hydrogel via Schiff-based chemically reaction. and is subsequently cross-linked under irradiation with UV light. The MOHA/MDM hydrogel attributes and its injectability, rheological, biomechanical, and swelling performances have been evaluated. The following changes are suggested prior to publication:

1-      The language of abstract should be edited to improve accessibility to a broad audience and the impact of findings. They should summarize their aim, objectives, and the main achievements in the abstract sections.

2-      Similarly, the introduction should be revised to improve clarity and more clearly highlight the research gap as it applied to intended applications.

3-      It should be mentioned about some of recently works close to your currently activities in introduction section and compare your findings with them in conclusion section.

4-       Line 173 to 214 should be move to introduction section because its general descriptions that are not relate to your finding.

5-      There are several aspects about physico-chemical characterization and hydrogel performances that need to be discussed in details.

Author Response

Detailed Responses to Reviewer 2:

The manuscript described the synthesize and characterization of double cross-linked injectable MOHA/MDM hydrogel via Schiff-based chemically reaction. and is subsequently cross-linked under irradiation with UV light. The MOHA/MDM hydrogel attributes and its injectability, rheological, biomechanical, and swelling performances have been evaluated. The following changes are suggested prior to publication:

Response: Thank you for your professional comments. We have amended our work according to your useful suggestion, thus resulting in a considerable improvement of our manuscript. And the revised text was highlighted in red.

1-The language of abstract should be edited to improve accessibility to a broad audience and the impact of findings. They should summarize their aim, objectives, and the main achievements in the abstract sections.

Response: Thank you for your useful suggestion. We have revised our abstract according to your professional comment. The following text is the updated abstract:

Injectable hydrogels possess tremendous merits for use in muscle regeneration; however, they still lack intrinsic biological cues (such as the proliferation and differentiation of myogenic cells), thus considerably restricting their potential for therapeutic use. Herein, we developed a double cross-linked injectable hydrogel, composed of methacrylamidated oxidized hyaluronic acid (MOHA) and muscular decellularized matrix (MDM). The chemical composition of the hydrogel was confirmed using 1H NMR and Fourier Transform infrared spectroscopy. To achieve crosslinking, the aldehyde groups in MOHA were initially reacted with the amino groups in MDM through a Schiff-based reaction. This relatively weak crosslinking provided the MOHA/MDM hydrogel with satisfactory injectability. Furthermore, the methacrylation of MOHA facilitated a second crosslinking mechanism via UV irradiation, resulting in improved gelation ability, biomechanical properties, and swelling performance. When C2C12 myogenic cells were loaded into the hydrogel, our results showed that the addition of MDM significantly enhanced myoblast proliferation compared to the MOHA hydrogel, as demonstrated by live/dead staining and Cell Counting Kit-8 assay after 7 days of in vitro cultivation. In addition, gene expression analysis using quantitative polymerase chain reaction indicated that the MOHA/MDM hydrogel promoted myogenic differentiation of C2C12 cells more effectively than the MOHA hydrogel, as evidenced by elevated expression levels of myogenin, troponin T, and MHC in the MOHA/MDM hydrogel group. Moreover, after 4–8 weeks of implantation in a full-thickness abdominal wall defect model, the MOHA/MDM hydrogel could promote the reconstruction and repair of functional skeletal muscle tissue with enhanced tetanic force and tensile strength. This study provides a new double-cross-linked injectable hydrogel for use in muscular tissue engineering.

Please note the revised text was highlighted in red.

2-Similarly, the introduction should be revised to improve clarity and more clearly highlight the research gap as it applied to intended applications.

Response: Thank you for your useful suggestion. We have revised our introduction according to your professional comment. Please note the revised text was highlighted in red.

3-It should be mentioned about some of recently works close to your currently activities in introduction section and compare your findings with them in conclusion section.

Response: Thank you for your useful suggestion. We have added recently works close to our currently activities in introduction section and compare our findings with them in conclusion section. Please note the revised text was highlighted in red.

4-Line 173 to 214 should be move to introduction section because its general descriptions that are not relate to your finding.

Response: Thank you for your useful suggestion. We have moved Line 173 to 214 should be move to introduction section according to your kindly suggestion. Please note the revised text was highlighted in red.

5-There are several aspects about physico-chemical characterization and hydrogel performances that need to be discussed in details.

Response: Thank you for your comment. We have added the discussion in terms of the physico-chemical characterization and hydrogel performances according to your kindly suggestion. Please note the revised text was highlighted in red.

Reviewer 3 Report

The authors prepared a new type of double cross-linked hydrogels utilizing hyaluronic acid and decellularized muscular matrix. The topic is interesting and I am quite confident that it will attract the attention of a wide readership. The major part of the manuscript is related to biological functionality which is unfortunately completely out of my competences, thus my comments are only related to chemistry/materials science aspects, in which I experienced several (major) shortcoming which must be improved before publishing.

1. I did not find a "Materials" section, so it is not clear which type of hyaluronic acid is used, how it is purified (if any purification was done) and what counterions are present. Even more importantly, a molar mass (range) must be given, based on either a data sheet or measurements by the authors.

2. Chemical aspects are rather superficial. Please explain the reaction resulting in methacrylated HA, I seriuosly doubt that the reaction proceeds as it is suggested in Fig. 1 and the text. Why would amide bonds form? Later, the reaction of amines and aldehydes result in Shiff bases (which I accept), but for amides, somehow both the aldehyde and methacyrlate should lose a hydrogen which does not make sense to me. Furthermore, nothing is given for the type and purity of methacrylate reagent.

3. Please add experimental evidence of Schiff-base formation. Do you have rheological data for this reaction? The change of dynamic moduli during UV-curing (Fig. 3) is almost negligible. Please explain. What would be the conversion of UV-crosslinking reaction?

4. Please compare shear moduli (Fig. 3B) and Young's modulus (Fig. 3C). It is unrealistic that difference with several orders of magnitude exists between them. I suspect errors in calculation for at least one of the moduli. Please explain and correct where needed.

5. Are these gels really injectable? What viscosity value proves this? Please add explanation.

Minor

Please add title for Y axis of Fig. 3D 

Overall, I recommend the major revision of the work.

Author Response

Detailed Responses to Reviewer 3:

The authors prepared a new type of double cross-linked hydrogels utilizing hyaluronic acid and decellularized muscular matrix. The topic is interesting and I am quite confident that it will attract the attention of a wide readership. The major part of the manuscript is related to biological functionality which is unfortunately completely out of my competences, thus my comments are only related to chemistry/materials science aspects, in which I experienced several (major) shortcoming which must be improved before publishing.

Response: Thank you for your professional comments. We have amended our work according to your useful suggestion, thus resulting in a considerable improvement of our manuscript. And the revised text was highlighted in red.

  1. I did not find a "Materials" section, so it is not clear which type of hyaluronic acid is used, how it is purified (if any purification was done) and what counterions are present. Even more importantly, a molar mass (range) must be given, based on either a data sheet or measurements by the authors.

Response: Thank you for your comment. The hyaluronic acid we used is medical grade and was purchased from Sigma-Aldrich (No. 924474, St. Louis, MO, USA). Its impurity <10 CFU/g Bioburden or <100 EU/g Endotoxin.

We have added their molecular weights of 490 kD in the revised manuscript. Please note the revised text was highlighted in red.

  1. Chemical aspects are rather superficial. Please explain the reaction resulting in methacrylated HA, I seriously doubt that the reaction proceeds as it is suggested in Fig. 1 and the text. Why would amide bonds form? Later, the reaction of amines and aldehydes result in Shiff bases (which I accept), but for amides, somehow both the aldehyde and methacyrlate should lose a hydrogen which does not make sense to me. Furthermore, nothing is given for the type and purity of methacrylate reagent.

Response: Thank you for your careful observation. We feel sorry for the imprecise description. In fact, the OHA was reacted with methacrylic anhydride. The information for methacrylic anhydride were listed as following: 1) Molecular weight: 154.16, 2) purity: > 94%, 3) Cas: No. 760-93-0, 4) purchased from Sigma-Aldrich (St. Louis, MO, USA). We have added these information in the revised manuscript. Please note the revised text was highlighted in red. And we have also corrected it in the revised figure 1 of our updated version.

For amides, the chemical equation for this reaction were displayed as the following figure:

  1. Please add experimental evidence of Schiff-base formation. Do you have rheological data for this reaction? The change of dynamic moduli during UV-curing (Fig. 3) is almost negligible. Please explain. What would be the conversion of UV-crosslinking reaction?

Response: Thank you for your professional comment and kindly suggestion. Considering our rheometer cannot measure while illuminating light in real-time, so we added a control group of MOHA/MDM without UV irritation. Consequently, the rheological experiments were carried out on three distinct groups: MOHA/MDM with UV irritation (referred to as MOHA/MDM + UV), MOHA with UV irritation (referred to as MOHA + UV), and MOHA/MDM without UV irritation (referred to as MOHA/MDM). And the updated results were displayed in the revised Figure 3 of our new version (shown in the following figure). Our data revealed that the levels of viscosity, G', and Young's modulus in hydrogels present a trend of MOHA/MDM + UV > MOHA +UV > MOHA/MDM, corroborating the two-step (Schiff-based reaction + UV irradiation) crosslinking mentioned in the study.

Figure caption: Characterizations of MOHA/MDM hydrogel. Viscosity depending on shear rate in MOHA and MOHA/MDM hydrogels with UV light at varying times and MOHA/MDM hydrogel without UV irritation (A). Dynamic storage modulus (G') of MOHA and MOHA/MDM hydrogels with UV light at varying times and MOHA/MDM hydrogel without UV irritation (B). Young's modulus of MOHA and MOHA/MDM hydrogels after 3 min UV irradiation and MOHA/MDM hydrogel without UV irritation (C). Swelling ratio of MOHA and MOHA/MDM hydrogels after 3 min UV irradiation and MOHA/MDM hydrogel without UV irritation from 0-24 h (D). *P < 0.05.

  1. Please compare shear moduli (Fig. 3B) and Young's modulus (Fig. 3C). It is unrealistic that difference with several orders of magnitude exists between them. I suspect errors in calculation for at least one of the moduli. Please explain and correct where needed.

Response: Thank you for your careful observation. We feel sorry for this error. We have reconducted this experiment for at least 3 times to achieve the new data. And we have corrected it in the revised Figure 3 of our new version.

  1. Are these gels really injectable? What viscosity value proves this? Please add explanation.

Response: Thank you for your careful observation. The rationale behind the injectability of our MOHA/MDM hydrogel can be summarized as follows:

1) The Schiff-base crosslinking in our hydrogel is relatively weak, which has been extensively demonstrated to enable injectability.

2) The injectability of our MOHA/MDM hydrogel was achieved prior to UV irradiation.

3) Our in vitro experiments provided evidence supporting the easy injectability of the MOHA/MDM hydrogel before UV irradiation. We observed that it could be easily injected through needles, forming a continuous line (Fig. 2C).

4) To further confirm the injectability of the MOHA/MDM hydrogel, we conducted low-high strain sweeps, applying a low strain of 1% and a high strain of 300%. Our data revealed that the loss modulus (G") was lower than the storage modulus (G') at low shear, while the G" was higher than G' at high shear (as shown in the following figure). These results provide further evidence of the satisfactory injectability of our MOHA/MDM hydrogel.

Figure caption: Alternative low-high strain sweeps of MOHA/MDM hydrogel at a low strain of 1 % and high strain of 300 %.

Minor

Please add title for Y axis of Fig. 3D

Response: Thank you for your careful observation. We have added the title for Y axis of Fig. 3D.

Overall, I recommend the major revision of the work.

Reviewer 4 Report

The authors thoroughly investigated a new injectable hydrogel in terms of mechanical properties (e.g., viscoelastic parameters) and proliferation of myoblasts in hydrogels, along with in vivo muscle repair evaluation on two groups of rats. The authors started crafting the basic biomacromolecule (Hyaluronic Acid, HA) by oxidizing HA (OHA) and adding methacrylate to prepare MOHA. They created a double network hydrogel by adding a muscular decellularized matrix (MDM), which possesses amino groups, and a photoinitiator. Although the procedure is laborious and time-consuming (thus, not ideal for large-scale application), the work is interesting and effective in regenerating muscular tissue.

The submission can be published after a minor revision once the authors address the following issues. 

An unclear method in Rheological tests

In Figure 3B, the intervals of the UV reaction kinetics are around 7,5 sec each. However, Figure 2D shows that such action is impossible to apply within this time frame. The author may have used a method/type of in situ UV-curing of the hydrogel. This should be clarified.

Also, In Figure 3B, there is no need for three orders of magnitude in the YY' axis just to fit the legend. Thus, the lower value of YY' could easily be 100 Pa, and the kinetics data will show clearly the details. 

Minor changes are necessary at particular points, that is:

1.) Be consistent with the expression <<Schiff-based>>, e.g., use Schiff-based in Figure 1; line 202: Schiff-based instead of "Schiff base" and throughout the text.

2.) Line 100: omit the connective dash in (25mm-diameter), as (25mm diameter)

3.) Line 161: define samples in the subsection "2.8. Biomechanical analysis", i.e., with/without cells, at which time of the preparation, etc.

4.) Page 6, lines 218 & 222, correct the Fig. 1A and 1B to Fig. 2A and 2B

5.) Line 281: separate the (G', B) to differentiate the rheological parameter and the lettering of the subplot.

Author Response

Detailed Responses to Reviewer 4:

The authors thoroughly investigated a new injectable hydrogel in terms of mechanical properties (e.g., viscoelastic parameters) and proliferation of myoblasts in hydrogels, along with in vivo muscle repair evaluation on two groups of rats. The authors started crafting the basic biomacromolecule (Hyaluronic Acid, HA) by oxidizing HA (OHA) and adding methacrylate to prepare MOHA. They created a double network hydrogel by adding a muscular decellularized matrix (MDM), which possesses amino groups, and a photoinitiator. Although the procedure is laborious and time-consuming (thus, not ideal for large-scale application), the work is interesting and effective in regenerating muscular tissue.

Response: Thank you for your professional comments. We have amended our work according to your useful suggestion, thus resulting in a considerable improvement of our manuscript. And the revised text was highlighted in red.

The submission can be published after a minor revision once the authors address the following issues.

-An unclear method in Rheological tests

Response: Thank you for your careful observation. We have added the explanation for the method of Rheological tests as flowing text:

The rheological experiments were carried out on three distinct groups: MOHA/MDM with UV irritation (referred to as MOHA/MDM + UV), MOHA with UV irritation (referred to as MOHA + UV), and MOHA/MDM without UV irritation (referred to as MOHA/MDM). To measure the viscosity of the hydrogel, an Anton Paar MCR302 rheometer with parallel-plate (25mm diameter) configuration was employed. In brief, the hydrogels were applied to cover the bottom plate of the rheometer with parallel-plate geometry. Subsequently, the gap between the plates was set at 1 mm, and the rheometer was covered. The hydrogel viscosity was then measured at room temperature across shear rates ranging from 0.1 to 100 s-1, both with and without UV irritation. Following exposure to UV irradiation (365nm, 10 mW/cm2) for 100 seconds or without exposure, dynamic rheology tests were conducted. These tests were performed at a fixed frequency of 6 rad/s and a strain of 1%, with the purpose of determining the storage modulus (G') of the hydrogels.

Please note the revised text was highlighted in red.

In Figure 3B, the intervals of the UV reaction kinetics are around 7.5 sec each. However, Figure 2D shows that such action is impossible to apply within this time frame. The author may have used a method/type of in situ UV-curing of the hydrogel. This should be clarified.

Response: Thank you for your careful observation. In Figure 2D, we used a small bottle to illustrate the gelation of MOHA/MDM and MOHA hydrogels via UV irradiation. However, in the rheological experiments, we use a in situ UV-curing method.

Also, In Figure 3B, there is no need for three orders of magnitude in the YY' axis just to fit the legend. Thus, the lower value of YY' could easily be 100 Pa, and the kinetics data will show clearly the details.

Response: Thank you for your careful observation and useful suggestion. We have zoomed the y-scale in the graphs for Figures 3a and b, thus making the data showed clearer.

-Minor changes are necessary at particular points, that is:

1.) Be consistent with the expression <<Schiff-based>>, e.g., use Schiff-based in Figure 1; line 202: Schiff-based instead of "Schiff base" and throughout the text.

Response: Thank you for your careful observation. We have unified expression of Schiff-based reaction throughout the text. Please note the revised text was highlighted in red.

2.) Line 100: omit the connective dash in (25mm-diameter), as (25mm diameter)

Response: Thank you for your careful observation. We have corrected the “25mm-diameter” as “25mm diameter” in the revised version. Please note the revised text was highlighted in red.

3.) Line 161: define samples in the subsection "2.8. Biomechanical analysis", i.e., with/without cells, at which time of the preparation, etc.

Response: Thank you for your careful observation. We have defined the sample as: “repaired samples at 4 and 8 weeks after implantation in the revised version”. Please note the revised text was highlighted in red.

4.) Page 6, lines 218 & 222, correct the Fig. 1A and 1B to Fig. 2A and 2B

Response: Thank you for your careful observation. We have corrected these errors. Please note the revised text was highlighted in red.

5.) Line 281: separate the (G', B) to differentiate the rheological parameter and the lettering of the subplot.

Response: Thank you for your careful observation. We have corrected this error. Please note the revised text was highlighted in red.

Round 2

Reviewer 2 Report

Dear Editor,

The authors have accurately implemented the comments of reviewers into the manuscript. I think that the submitted manuscript would be able to accept and publish in Materials without more modification and corrections.

Best

Reviewer 3 Report

The authors really improved the quality. Nevertheless, I am still a chemist, thus I would recommend acceptance after a proper biologist approves those parts.